# Spin Orbit Coupling in Orthogonal Charge Transfer States: (TD-)DFT of Pyrene—Dimethylaniline

**DOI:** 10.3390/molecules27030891

**Published:** 2022-01-28

**Authors:** Shivan Bissesar, Davita M. E. van Raamsdonk, Dáire J. Gibbons, René M. Williams

**Affiliations:** Molecular Photonics Group, van ’t Hoff Institute for Molecular Sciences (HIMS), Universiteit van Amsterdam, Science Park 904, 1098 XH Amsterdam, The Netherlands; shivan_b@live.com (S.B.); davita.vanraamsdonk@student.uva.nl (D.M.E.v.R.); d.j.gibbons@uva.nl (D.J.G.)

**Keywords:** triplet formation, charge recombination, charge separation, intersystem crossing, SOCT-ISC, SOCME, electronic coupling

## Abstract

The conformational dependence of the matrix element for spin–orbit coupling and of the electronic coupling for charge separation are determined for an electron donor–acceptor system containing a pyrene acceptor and a dimethylaniline donor. Different kinetic and energetic aspects that play a role in the spin–orbit charge transfer intersystem crossing (SOCT-ISC) mechanism are discussed. This includes parameters related to initial charge separation and the charge recombination pathways using the Classical Marcus Theory of electron transfer. The spin–orbit coupling, which plays a significant role in charge recombination to the triplet state, can be probed by (TD)-DFT, using the latter as a tool to understand and predict the SOCT-ISC mechanism. The matrix elements for spin–orbit coupling for acetone and 4-thio-thymine are used for benchmarking. (Time Dependent-) Density Functional Theory (DFT and TD-DFT) calculations are applied using the quantum chemical program Amsterdam Density Functional (ADF).

## 1. Introduction

Using computational chemistry to determine the matrix element for spin–orbit coupling [1,2] (SOCME) in simple organic molecules like acetone, is a challenging task [3]. SOCME determination in charge transfer systems [4] goes beyond this, since the transition from a charge transfer state to a local triplet excited state falls within the Marcus theory [5,6] of electron transfer, and can occur in large bifunctional molecules containing an electron donor and an electron acceptor [7,8]. This process, triplet charge recombination (TCR), plays a role in organic photovoltaic materials, heavy-atom-free photosensitizers, as well as in LEDs. “Triplet formation by charge recombination is a phenomenon that is encountered in many fields of the photo-sciences and can be a detrimental unwanted side effect, but can also be exploited as a useful triplet generation method, for instance in photodynamic therapy” [9]. It can also be of importance in a great variety of other applications such as in organic photocatalysis and solar energy harvesting [10,11,12]. Photosensitizers often contain transition metals [13] such as Ru, [14] Pd [15] and Pt [16,17]. In these complexes, intersystem crossing (ISC) is efficient due to spin–orbit interactions, [18] a relativistic effect usually present in atoms with large nuclei, commonly called the heavy-atom effect. The corresponding ISC is known as spin–orbit intersystem crossing (SO-ISC) [19]. Introducing heavy atoms to form triplet states is far from ideal: increasing cost, low solubility, and not to forget, their environmental impact is significant. The use of heavy-atom-free triplet sensitizing dyes is an emerging research field. However, it is still difficult to design these structures with efficient ISC due to a lack of understanding of the relationship between ISC and the molecular structure.

To be able to design efficient heavy-atom-free triplet sensitizer molecular systems, knowledge about ISC in these structures should be enhanced. An approach that has been shown to efficiently produce triplet states is by using charge recombination (CR) from a charge transfer (CT) state. In this process, a large orbital angular momentum change is induced by the CR, which can now compensate for the electron spin-flip, to satisfy the rule of angular momentum conservation. This ISC is called spin–orbit charge transfer ISC (SOCT-ISC) [20].

Already in 1963, El-Sayed predicted the basis of the SOCT-ISC via his triplet selection rules [21]. The rate of ISC becomes larger if a radiation-less transition is accompanied by a change of molecular orbital (MO) type, which physically implies that an increased spin–orbit coupling (SOC) between the ^1^CT and T_1_ state can result in higher ISC rates. Later, in 1981, the first molecular system undergoing this process was discovered by Okada and colleagues [22]. Thereafter, Van Willigen suggested that if CR occurs at two aromatic planes, in combination with a large dihedral angle, it can generate the torque needed to spin-flip the electron [23]. Based on this suggestion, Wasielewski et al. proposed that when the MOs are (nearly) perpendicular to each other, SOCT-ISC is favored over the alternative HFI-ISC (the latter falls outside of the scope of this work, HFI = Hyper Fine Interactions) [24]. Further research regarding this observation confirmed the relationship and also found that the polarity of the solvent plays an important role in the rate of ISC^22^. These observations show that the CT state, as well as the orthogonality of the MOs, is of crucial importance for the efficient production of triplets via the SOCT-ISC mechanism. Mataga et al. already stated that “the matrix element of the spin-orbit coupling (SOCME) of the CT state and triplet state is increased in the perpendicular orientation” [22].

The increased interest in this mechanism has not only affected experimental research, but also computational chemists. The use of computational tools to study chemistry problems is emerging, and the theory and usability of computational chemistry has reached a level that is advanced enough to study excited state charge transfer processes [9]. Time-dependent density functional theory [25] (TD-DFT) calculations can be performed to study the SOCT-ISC mechanism [26].

The aim of this work is to develop a method within ADF, using TD-DFT, to properly determine the matrix element of spin–orbit coupling for charge recombination in electron donor–acceptor molecules, as well as the energetics and conformational effects on this quantity. We will describe the various aspects that play a role in the decay of charge separated states into local triplet excited states by the SOCT-ISC mechanism for one particular molecule (N-methyl-N-phenyl-1-pyrene-methanamine). Thereby we generate a framework that can easily be applied to other systems.

For an overview of the SOCT-ISC mechanism, a few clear review papers have been published [9,19,20,27,28]. To generate triplets via the SOCT-ISC mechanism, three photophysical pathways are involved (see Figure 1): spin-allowed initial charge separation (CS) from S_n_ → ^1^CT, CR_S_ from the ^1^CT to the ground state (GS), and spin-forbidden CR from ^1^CT to T_1_ (CR_T_) [26]. The ultimate goal is to find an optimum between initial charge separation and recombination pathways as discussed by Buck et al. [26]. They stated in their studies that the spin allowed processes, charge separation (S_2_ → ^1^CT) and CR to the ground state (CR_S_), are dominant in the yield of the triplet formation [26]. In order to favor the SOCT-ISC mechanism, the initial CS and CR pathways should be fine-tuned [9] by optimizing the SOCME and the electronic coupling.

These aspects, that play a role in the SOCT-ISC mechanism, can be described via computational methods; however, to our knowledge, a complete clear overview and application of how to use DFT calculations to tackle this complex mechanism was not made before. This work will include the variation of the SOCME ($V_{SOC}$), the electronic coupling ($V_{CT}$) [29] and other parameters that are of importance for the formation of triplet states by CR as well as solvent effects (with the conductor-like screening model, COSMO) [30] that influence the efficiency of the SOCT-ISC mechanism. The radiative lifetime of the triplet state will also be included. The limitations of the computational program, Amsterdam Density Functional (ADF), will be discussed. The aim is to give an overview and application of how to use ADF to describe, understand and predict the SOCT-ISC mechanism.

Within this study, a prototypical molecule, N-methyl-N-phenyl-1-pyrene-methanamine (PyrDMA, see Figure 2), is studied, that shows sub-ns charge recombination to the triplet state, as a model system to calculate the various aspects that play a role in the spin–orbit charge transfer-intersystem crossing (SOCT-ISC) mechanism. PyrDMA is a relatively small molecule, with a short flexible spacer, allowing for the computational study of dihedral angular effects on the SOCT-ISC mechanism.

The experimentally measured triplet charge recombination rates of PyrDMA in different solvents are indicated in Table 1.

The PyrDMA system, containing the text-book chromophore pyrene [31] and one of the most standard electron donors, dimethylaniline, [31,32,33,34] shows one of the fastest rates of triplet formation by charge recombination ever measured. Sub-ns charge recombination to the triplet is experimentally observed in n-hexane. The rates for triplet charge recombination vary strongly in the solvents. The charge separation process in PyrDMA and similar molecules occurs on a 1 ps timescale (see later, section on energetics). We set out to quantitatively explain this behavior and make a roadmap for designing systems with a high triplet generation yield via the SOCT-ISC mechanism.

In this work a nomenclature for the excited states is used that is consistent and directly related to the outcome of the TD-DFT calculations. This often implies that the S_1_ state equals the singlet charge transfer state (^1^CT, the lowest state with singlet character of the system). CT character of states will always be specified, as far as possible. The triplet charge transfer state (^3^CT = T_2_) does not play a role in the photophysics of PyrDMA [22]. As also discussed in our previous work, [9] the conversion of the ^1^CT into the ^3^CT is expected to occur on a 10 ns timescale. However, computational results regarding the ^3^CT are presented in our manuscript when appropriate.

## 2. Experimental: Computational Methods

Structures were optimized and properties were calculated with the ADF DFT package (SCM, version 2019), using the SCF convergence criterion (1 × 10^−6^) in ADF, on the Lisa cluster of SURFsara [35,36]. As a starting structure PyrDMA was built and optimized in SPARTAN [37]. All geometry optimizations were conducted using the zero-order regular approximation (ZORA) for relativistic effects [38]. ADF basis sets of triple zeta plus polarization (TZP or TZ2P) were used (as specified), with no frozen core. They both gave consistent results. Representative ADF input and output files are provided as well as an extensive description of the procedure with (repeated) references (see Appendix A). Coordinate files for the various excited states are supplied as mol2 files. The results were visualized with UCSF Chimera [39].

## 3. Results and Discussion

### 3.1. Benchmarking the SOCME

Transitions between singlet and triplet states are forbidden in a non-relativistic framework; however, intersystem crossing from a singlet to a triplet state is possible in the presence of spin–orbit coupling. It is challenging to calculate the SOCME factor in both an effective and accurate way. Using quantum chemistry, it is possible to gain more insight into the SOCME, because it is the result of the spin-operator expressed by its Hamiltonian [9,40]. This spin–orbit Hamiltonian describes the interaction between the spin and orbital motions of an electron and induces singlet and triplet excitations. The coupling of the spin and orbital momenta of the nucleus and electron is described by the Hamiltonian:(1)$${\hat{H}}_{SO}=\alpha_{fs}^{\;\;2}\overset{N}{\underset{\mu }{\sum }}\overset{n}{\underset{m}{\sum }}\frac{z_{\mu }}{r_{m\mu }^{3}}\vec{L_{m}}\vec{S_{m}}$$

$\alpha_{fs}$ is the fine structure constant. $z_{\mu }$ represents the effective nuclear charge for nucleus μ. $\vec{L}$ is the orbital momentum and $\vec{S}$ is the spin momentum operator. The distance between the nucleus and electrons is represented by $r_{m\mu }$. This is an effective one-electron operator. The SOC matrix element [41,42] was computed by considering the three degenerate T_1_ triplet states (m=0,±1).
(2)$$V_{SOC}=\langle S_{1}|{\hat{H}}_{SO}|T_{1}\rangle =\sqrt{\Sigma_{m=0,\pm 1}{\left\langle S_{1}|{\hat{H}}_{SO}|T_{1}^{m}\right\rangle }^{2}}$$

Calculating the SOCME between the ^1^CT and T_1_ state can be performed by using TD-DFT. The SOCME is represented as the waveform root-mean-square average of the three sublevels, obtained by applying the SOPERT keyword in ADF (See Appendix A).

In order to test that our method (ADF), the functionals and basis sets are appropriate, it is important to perform benchmark calculations [3]. The first step in this work was therefore to benchmark the method for the SOC matrix ($V_{SOC}$) determination, for which acetone and 4-thio-thymine were used as reference molecules [3]. The SOCME ($V_{SOC}$) for these molecules was calculated within our work with different exchange correlation functionals (see Table 2). Using the ground state geometry resulted in much lower values for $V_{SOC}$ and, therefore, the calculations were done at the optimized excited state S_1_ geometry (PBE exchange correlation/TZP basis). The SOCME values were benchmarked with ADF using different exchange functionals. We have mainly selected the higher S_1_-T_2,_ values from the output (Table 2). The SOC matrices that we obtained correlate well with those previously reported in a benchmark study [3].

The $V_{SOC}$ literature values for 4-thio-thymine lie within the range 138–206 cm^−1^, while the values for acetone lie within 44–88 cm^−1^, depending on different computational approaches, exchange correlation (XC) functionals and basis sets [3].

The exchange correlation functionals resulted in $V_{SOC}$ values of 61.97 ± 2.24 cm^−1^ for acetone and 156.62 ± 21.67 cm^−1^ for 4-thio-thymine (see Table 2). The exchange correlation functional didn’t significantly affect the SOCME value, as expected. All $V_{SOC}$ values, except for the one which used BhandH, lay in the range from the literature. It can be noted that the highest values were found for the LB94 exchange correlation and, furthermore, this benchmark study was done relative to the reported work on other computational methods [3]. It should be noted that we do not claim comparison to precise experimental data that is unknown to us. Furthermore, we applied a relatively simplistic approach in which the effects that the different XC functionals may have on the optimized structures were not taken into account.

In range-separated hybrid (RSH) DFT methods, the amount of exact exchange increases with the electron–electron distance. In many RSH approximations, this is used to restore the correct long-range asymptotic behavior of the corresponding potential [43]. Their proper asymptotic behavior renders RSH methods potentially more accurate for the description of electronic excitations. The RSH functionals CAM-B3LYP and the CAMY-B3LYP functionals correctly predicted the SOCME values and also predicted the correct matrix element $V_{SOC}$ with the state shift in 4-thio-thymine. This is especially important for describing CT states [44]. The CAM-B3LYP, however, cannot be used in geometry optimizations with ADF. Therefore, the calculations for the geometry optimizations were performed using TD-DFT with the range-separated CAMY-B3LYP functional (next to the PBE functional, that was also used). The CAMY-B3LYP functional made use of a range separation parameter γ, which was the inverse distance and measures how fast range separation switches from short- to long-range. Changing the range separation parameter did not significantly affect the $V_{SOC}$ values for acetone and 4-thio-thymine (see Figure 3). For PyrDMA, choosing a low range separation parameter led to a higher $V_{SOC}$ value, thus, values 0.34 and 0.1 were used. The former is the default value for the range separated CAMY-B3LYP hybrid (see Figure 3).

### 3.2. Excited State Geometries of PyrDMA

After benchmarking the SOCME method and assessing the correct XC for charge transfer systems, the next step, in order to study the SOCT-ISC mechanism, was to conduct geometry optimizations. Geometry optimizations were performed for the ground-state (GS or S_0_), and the singlet (S_n_) and triplet (T_n_) excited states of PyrDMA. The intersystem crossing rate depends rather strongly upon the mutual configuration of donor and acceptor groups, as well as the solvent polarity [45]. Therefore, the optimized geometries of the excited states were calculated in the gas-phase (Gas), n-hexane (NHX) and acetonitrile (ACN). Representative geometries are presented in Figure 4, Figure 5 and Figure 6.

It can be noted that T_2_ (=triplet charge transfer state) does not play a role in the photo-physics (see also Section 3.3). Furthermore, the strongest structural deviation of the triplet state (T_1_) is remarkable (see Figure 5).

The comparisons between the optimized geometries showed a strong structural similarity between the S_1_ (^1^CT) and the T_2_ (^3^CT) structures (see Figure 4), which makes physical sense as the S_1_ and T_2_ are almost isoenergetic. The arrangement of the atoms around the nitrogen atom in the S_1_ state was more flat, compared to the ground state S_0_, in which the nitrogen atom was more pyramidalized. The optimized geometries formed a “waving hand” structure (see Figure 5), which moved up and down with the excited states. Energetic and structural information of the various excited states and their geometries is represented in Table 3, Table 4 and Table 5.

In Table 3 and Table 4, information on the total bonding energy at the optimized geometries is represented, as output energies, as well as the energy difference relative to the ground state. Total bonding energy does not take into account excitation energies, but consists of several contributions: electrostatic energy, kinetic energy, Coulomb (steric + orbital interaction) energy, XC energy and solvation (see ADF manual for further information). Optimized ground and excited state geometries vary with the solvent, but the total bonding energy difference between, e.g., the ground state and the local pyrene triplet state (T_1_) is consistent. In Table 5, we indeed observe the “up-down” movement of the excited states to form the molecular “hand wave”.

Following the El Sayed rules, the ISC rate becomes higher if a radiation-less transition is accompanied by a change of molecular orbital (MO) type, as previously mentioned. Thus, an increased spin–orbit coupling (SOCME) between the ^1^CT and T_1_ state results in higher ISC rates. When the donor and acceptor are in an orthogonal orientation, there will be a change in orbital angular momentum, which will induce a large SOCME, allowing for CR_T_. Studying the HOMO (−) and LUMO (+) MOs can give insight into whether or not a large SOCME can be expected for the system. Multiple MOs may contribute equally to an excited state and can be transformed to a Natural Transition Orbital (NTO) as shown in Figure 7. With the NTO, the qualitative description of the electronic transition can be simplified and it gives insight into the localization of excitations.

The NTOs [46] show a clear charge transfer transition between the HONTO of the aniline (red-blue) to the LUNTO on the pyrene (brown-cyan). There is a strong similarity between the singlet and triplet charge transfer state. The energy difference (13 meV) indicates a *J* value of 6.5 meV.

### 3.3. Excitation Energies and Scheme at Optimized Geometries

The excitation energies increase in the order S_0_, T_1_,T_2_, S_1_ and S_2_ (see also Section 3.6). From the order of excitation energies of PyrDMA, the following conclusions can be drawn about the states:

S_2_ is the locally excited (LE) state of pyrene, from which charge transfer occurs. The numerical results match quite well with the experimentally measured singlet excited state of pyrene $E_{S_{2}}=3.26$ eV (see also Section 3.6).

S_1_ state is the charged separated singlet charge transfer state ^1^CT. The S_1_ state has an excitation energy near $E_{S_{1}}=2.87$ eV.

T_2_ state is the triplet charge transfer state ^3^CT, which has about the same excitation energy as the singlet charge transfer state. The triplet charge transfer state does not play a role in the photophysical processes, since the lifetime of the ^1^CT state is only 40 ps, as determined experimentally. The proton hyperfine interactions proceed with a rate of about 8 × 10^7^ s^−1^ [9].

T_1_ is the final triplet product state, which can again decay to the ground state S_0_ [47]. The excitation energy of the triplet state matches well with the experimental triplet excitation of pyrene. For a better overview of the scheme, see see also Section 3.6.

### 3.4. Angular Dependence of Spin–Orbit Coupling and of the Electronic Coupling

The electronic coupling [48] and SOCME values [49] need to be calculated at different nuclear coordinates [50] in order to determine their angular dependence. The dihedral angle defined in Figure 8 is the nuclear coordinate which is varied in this study.

Different parameters are needed for the Marcus theory and the electronic coupling is one of them. The electronic coupling facilitates charge separation (CS). CS can be visualized by studying the molecular orbitals and their electronic transitions. For example, the electronic transition from a frontier HOMO on the donor-unit to the LUMO of the acceptor molecule results in a charge-separated state [51]. The electronic coupling is the same as the effective charge transfer integral ($V_{CT}$) [52]. The following equation shows the correlation of the charge transfer integral with the transfer integral J, overlap integral S, and site energy ε.
(3)$$V_{CT}=\frac{J-\left(\frac{s\left(\varepsilon 1+\varepsilon 2\right)}{2}\right)}{1-s^{2}}$$

The charge transfer’s integral corresponding to the electronic coupling between the LE state and the CT-state can be calculated with the quantum chemical program ADF. The charge transfer integrals are calculated using a fragment approach (see Figure 9). The fragment approach makes use of MOs on the individual fragmental molecules as a basis set of the calculations on a system containing two or more fragment molecules [53]. The aniline donor and pyrene acceptor are split and form two fragments. The CH_2_ group is deleted, and hydrogen atoms are added to the free valences (the dangling bonds are passivated with H-atoms). The two fragments (pyrene and N-H-methylaniline) are first computed separately, followed by a computation that calculates the interaction between the two fragments. The charge transfer’s integrals, site energies, and overlap integrals between the two fragment orbitals are calculated. The electronic coupling is calculated as a function of the dihedral angle between pyrene and N-H-methylaniline (see Figure 9) at the geometry of the precursor state, in this case the S_2_.

Figure 9 clearly shows that optimal charge transfer interaction can occur at ~180°. Around 0° and 360° the $V_{CT}$ values also rise (data not shown), but due to interactions between the N-methyl group and a H atom of pyrene, steric repulsive interactions dominate (see total bonding energies). The 45 to 300 degrees range has been chosen for clarity. The E_0_ shows that the preferred conformation in the ground state is characterized by a dihedral angle of about 70° (for the other solvents: see Appendix A).

The SOCME ($V_{SOC}$) values are required for the charge recombination rates in the SOCT-ISC mechanism. Studying an interaction between two states is normally studied from the precursor state, which implies that the SOCME values should be calculated at the S_1_ geometry. From a Potential Energy Surface Scan (PESScan), the SOCME values of the singlet charge transfer state (S_1_-T_1_ matrix element) were computed as a function of the dihedral angle of the optimized S_1_ geometry in different solvents. (See Figure 10).

In Table 6, the values of the electronic coupling, the spin–orbit coupling at optimized geometries with this method as well as at their maxima, are given. This gives an overview of the range of angles and couplings that can be considered. The electronic coupling and the SOCME curves are combined in Figure 11, in which the squared quantities (proportional to the optimal rates) are displayed.

The singlet charge transfer state ^1^CT shows slower SOCT-ISC at dihedral angles near 180° in the solvents of ACN and NHX. The dihedral angle for optimal SOCT-ISC is around 90°. The SOCME values become enhanced in this near perpendicular configuration, which has previously been observed [22]. Mataga et al. stated that “the matrix element of the spin-orbit coupling (SOCME) of the CT state and triplet (T_n_) state is increased in the perpendicular orientation”. ACN is a highly polar solvent and thus has a more significant impact on the geometry of the dipolar charge transfer state S_1_, as compared to n-hexane and the gas-phase. This leads to a modulated SOCME curve for ACN. Furthermore, the ACN curve shows a discontinuous behavior, which can be explained by an electronic effect related to charge stabilization of the dipolar charge transfer state in ACN (see next section) and is correlated to a reduced distance between the charges.

The conformational dependence of the electronic coupling behaves complementary to the SOCME curves, which is in line with the current scientific paradigms (i.e., in line with the current understanding of the mechanism).

The electronic coupling near 180° in the solvents is high, which suggests that charge separation can occur very rapidly near this conformation. The orthogonal orientation of the π-type MOs between the donor and acceptor favors the SOCT-ISC mechanism, however, it is not favorable for electron transfer, due to the diminished electronic coupling in this configuration. This substantiates that the electronic coupling diminishes with a dihedral angle going towards orthogonality. At the optimized geometries, there is a lower charge separation and a higher charge recombination rate.

Even though the angular dependence of charge transfer integrals has been reported before, [54,55] we believe that the combination of calculating the spin–orbit coupling matrix element ($V_{SOC}$) and the electronic coupling ($V_{CT}$) for one molecule is unprecedented. It has to be highlighted that the total energies (E_0_) in Figure 9 and Figure 10 indicate that the conformation of PyrDMA is rather restricted and that, for instance, the range of angles between 120 and 190 degrees is totally inaccessible for the molecule. It has to be realized that the curves in Figure 9 and Figure 10 are hypothetical curves obtained by forcing the molecule in certain conformations that it normally would not or could not attain. This is the only way to determine the angular dependence of these quantities.

### 3.5. Angular Dependence of the Charge Transfer Character, Electron Hole Distance and Energetics

Next to the spin–orbit coupling matrix element ($V_{SOC}$) and the electronic coupling ($V_{CT}$), other properties as a function of the dihedral angular coordinate can be probed, like CT character and the distance between the centers of charge [56] (see Figure 12). The method for charge transfer descriptors is based on that of Plasser, Lischka et al. [57]. The NTO and DESCRIPTORS key words that allow for the calculation of $R_{HE}$, the average distance related to the electron-hole separation upon electronic excitation. A large distance can be found for CT excitations and a short distance for valence excitations. The charge transfer descriptor gives insight into the charge-transfer character, which has a value between 0 and 100%. A value of 0% represents a locally excited state with no CT, and 100% a totally charge separated state.

The singlet charge transfer states show near 98% charge transfer character in gas-phase and in n-hexane, while a lower charge transfer character is observed in ACN. The charge transfer state has, on average, a lower charge transfer of approximately 86% in ACN. The electron-hole distance is approximately constant near 6.3 Å. The electron-hole distance does not significantly change for the singlet and triplet charge transfer states in gas and n-hexane (see Appendix A).

The abrupt change in distance in ACN is correlated to the changes in the SOCME value as observed in Figure 13, which suggests that this discontinuity is not an ‘error’ [58]. The distance shows a significant change in ACN, which most likely is related to an electronic effect induced by coulombic charge stabilization of the charge transfer state in the highly polar ACN solvent, accompanied by a structural contraction. Measuring the center to center distance with the optimized geometries agrees with this interpretation, as a slightly contract structure is present in the ACN solvent (see Table 7). However, in our work we mainly focus on NHX as a solvent, in which fast CRT occurs. Therefore, we will not expand on this intriguing aspect in ACN. Furthermore, our solvent model (COSMO) may be too simplistic and explicit solvent models may be needed.

Figure 13 shows that, at a dihedral angle of ~95°, the SOCME and the electron-hole separation distance show a minimum, and both show a maximum at ~290°. The reduced $R_{HE}$ values correlate with a structural change and reduced distance between donor and acceptor.

### 3.6. Energetic and Kinetic Considerations for Charge Separation and Recombination Pathways

The previous sections describe the most important computational results of our study. In order to make a correlation to the experimental kinetic and energetic data, we now need to put these results within the framework of the Marcus theory of electron transfer. Therefore, in this section, we make an overview of the parameters that play a role in the energetic and kinetic aspects of the charge separation and recombination processes occurring in PyrDMA. We now combine experimental and computational data to give a coherent description of the PyrDMA system. Some important parameters are compiled in Table 8 and their determination is described thereafter. At the end of this section, we also discuss the problems we encountered with the Huang–Rhys factor, and correlate experimental rates with theoretical rates.

An important parameter from the Marcus theory [5,6] that can be calculated is the reorganization energy [59] (λ). The reorganization energy consists of external ($\lambda_{s}$) and internal ($\lambda_{i}$) components.
(4)$$\lambda =\lambda_{i}+\lambda_{s}$$

The focus is first on calculating $\lambda_{i}$, which relates to the change in energy associated with changes in nuclear coordinates upon the conversion of, for example, the neutral state to a charged state. The reorganization energy for the electron donor ($\lambda_{D}$) and the electron acceptor ($\lambda_{A}$) of the molecules can be calculated using the four-point method in ADF [60].
(5)$$\lambda_{int}=\lambda_{D}+\lambda_{A}$$
(6)$$2\lambda_{A}=\left(E_{0}^{-}-E_{-}^{-}\right)+\left(E_{-}^{0}-E_{0}^{0}\right)$$
(7)$$2\lambda_{D}=\left(E_{0}^{+}-E_{+}^{+}\right)+\left(E_{+}^{0}-E_{0}^{0}\right)$$

$E_{G}^{S}$ represents the energy of the state with charge S at geometry G [61]. For example, $E_{0}^{-}$ is the energy of the anion at the optimized geometry of the neutral molecule (see also the Appendix A).

The reorganization energy can be determined by calculating the different energies with the use of DFT. Four geometry optimizations need to be performed for the neutral fragments, the anionic, and cationic fragments (specifying the charges and unpaired electrons). Thereafter, four single point calculations need to be performed. The best model compounds that can be used for PyrDMA are N,N-dimethylaniline for the donor and 1-methylpyrene for the acceptor. They were used to estimate the internal reorganization energies. The acceptor contributes less (~0.13 eV) than the donor.

Calculated internal reorganization energies are presented in Table 8. Clearly, a relatively large internal reorganization energy is observed. The internal reorganization energy of dimethylaniline has been calculated before [62]. The normal mode analysis method resulted in 5400 and 3500 cm^−1^ (0.67 and 0.43 eV) giving an average of 0.55 eV. Other work [63] resulted in values of 0.41 and 0.28 eV (average of 0.345 eV) for N,N-dimethylaniline.

If we apply the distance reported in Table 7 for the different states, a slight variation is introduced (See Table 9).

The solvent reorganizational energy $\lambda_{s}$ is defined by Equation (8).
(8)$$\lambda_{s}=\frac{e^{2}}{4\pi \epsilon_{0}\;}\left(\left(\frac{1}{2r^{+}}+\frac{1}{2r^{-}}\right)-\frac{1}{R_{C}}\;\right)\left(\frac{1}{n^{2}}-\frac{1}{\epsilon_{s}}\right)$$

n here is the refractive index of the solvent. $\epsilon_{s}$ is the relative dielectric constant of the medium. $r^{+}$ and $r^{-}$ are the cation and anion radii [64]. $R_{C}$ is the center to center distance between pyrene and dimethylaniline. $\epsilon_{0}$ is the vacuum permittivity, *n* the refractive index and e the elementary electronic charge. The last parameter that is needed for the Marcus equation is the Gibbs energy (ΔG). The Gibbs free energy for charge separation can also be estimated using the following equation:(9)$$\Delta G_{CS}=e\left(E^{0}\left(D^{+}/D\right)-E^{0}\left(A/A^{-}\right)\right)-{\;}^{1}\Delta E_{00}-\frac{e^{2}}{4\pi \epsilon_{0}\epsilon_{S}R_{C}}$$

The Gibbs free energy consists of four terms: the energy it costs to oxidize the donor $E^{0}\left(D^{+}/D\right)$, the energy required to reduce the acceptor $E^{0}\left(A/A^{-}\right)$ and how much (useable) energy is put into the system by excitation ${\;}^{1}\Delta E_{00}$. This is the energy of the locally excited state S_2_. A Coulomb term, $\frac{e^{2}}{4\pi \epsilon_{0}\epsilon_{S}R_{C}}$, which describes solvent effects. The oxidation potential of the DMA donor, and the reduction potential of the Pyrene acceptor, as well as its singlet state energy in ACN are well known [64,65].

Polar solvents are better at stabilizing charged molecules, making charge separation more favorable. Changing to a different solvent requires a solvent correction term [32,33]:(10)$$\Delta G_{CS}=e\left(E^{0}\left(D^{+}/D\right)-E^{0}\left(A/A^{-}\right)\right)-{\;}^{1}E_{00}-\frac{e^{2}}{4\pi \epsilon_{0}\epsilon_{s}R_{C}}-\frac{e^{2}}{8\pi \epsilon_{0}}\left(\frac{1}{r^{+}}+\frac{1}{r^{-}}\right)\left(\frac{1}{\epsilon_{EC}}-\frac{1}{\epsilon_{s}}\right)$$

Using the Gibbs free energy for charge separation, the rates for charge separation can now be calculated. As observed in Table 8, the Gibbs free energy is highest in ACN and lowest in the gas phase, which implies that the driving force for charge separation is the highest in ACN.

The Gibbs energy equals ΔE and can also be calculated by taking the difference in energy of the two states of interest. The excitation energies, in this case, can be used to calculate the Gibbs energy (see Figure 14).

In Figure 14, the different energies of the states, as well as the couplings, are presented for the charge separation as well as for the charge recombination process for PyrDMA in NHX.

From these calculated numbers (see Section 3.3) presented in Figure 14, we can estimate a triplet emission at 640 nm, the Gibbs free energy changes for charge separation (−0.292 eV) and charge recombination to the triplet (−0.913 eV). The energetic level of the triplet excited state of the pyrene can be estimated from the phosphorescence of pyrene or substituted pyrene [65,66]. Emission maxima are reported at 647, 664 and 680 nm for the phosphorescence of substituted pyrene units. Thus, the triplet level is estimated to be between 1.91 and 1.82 eV. For pristine pyrene it is 2.1 eV, (2.08 eV for 1-methylpyrene). The energy of the singlet state of pyrene is reported to be 3.34 eV in nonpolar and 3.33 eV in a polar medium [67].

Classical Marcus Theory does not correctly predict the rates in the Marcus inverted regime [68]. The rates in this regime can be determined with the Marcus–Levich–Jortner (MLJ) theory to calculate the CS and CR rates. Therefore, we first set out to apply the single mode semi-classical Marcus equation (MLJ), [54] in which the electronic coupling is substituted by the spin orbit coupling: [42]
(11)$$k_{CR}=\frac{2\pi^{3/2}}{h\sqrt{\lambda_{s}k_{B}T}}{\left|V_{SOC}\right|}^{2}\sum_{n=0}^{m}\frac{e^{-S}S^{n}}{n!}\exp\left(\frac{-{\left(\Delta G+\lambda_{s}+nh\omega \right)}^{2}}{4\lambda_{s}k_{B}T}\right)$$
(12)$$S_{eff}=\underset{i}{\sum }S_{i}$$
(13)$$\omega_{eff}=\underset{i}{\sum }\frac{\omega_{i}S_{i}}{S_{eff}}$$
with $S_{i}=$ $\lambda_{i}/h\omega$. The factor $S_{eff}$ is the Huang–Rhys factor of these effective modes. It is a measure of the strength of electron–phonon coupling. With the individual Huang–Rhys factors $S_{i}$ for each vibrational mode, an effective Huang–Rhys factor can be calculated.$\;\omega_{eff}$ denotes an effective mode frequency, which can be calculated with the individual high frequency $\omega_{i}$ normal modes [69].

Interestingly, for a molecule very similar to PyrDMA, electron transfer parameters have been reported, using a very different (multi-parameter fitting) approach. For a ‘donor-inversed’ molecule (P1D), in which the amino unit is para relative to the CH_2_ linker, such parameters have been estimated ($\lambda_{i}\;$= 0.543 eV; S = 0.65, hω = 0.840 eV, ω = 6775 cm^−1^) [70]. The Gibbs free energy for electron transfer in acetonitrile was evaluated experimentally to be −0.48 eV, and is also reported in this work.

However, so far we were unable to make a good determination of S with DFT for PyrDMA. The S values obtained for PyrDMA so far were very low (too low, S~0.001), which we assume to be due to the large conformational changes between the charge transfer state and the triplet state. Based on the couplings, the energetic and kinetic experimental data, we estimate the S values for PyrDMA to be ~7, ~5, ~1 for CS, CRS and CRT, respectively (data not shown). Unfortunately, we cannot approach the rates with the Semi-Classical Marcus model.

Having determined the *V_SOC_* value (2.46 cm^−1^), as well as the solvent reorganization energy ($\lambda_{s}\;$= 0.00135 eV) in NHX at the optimized geometry, we can now estimate the charge recombination rate (*k_CR_*) with the Classical Marcus equation (see Table 10).
(14)$$k_{CR}=\frac{2\pi^{3/2}}{h\sqrt{\lambda k_{B}T}}{\left|V_{SOC}\right|}^{2}*\exp\left(\frac{-{\left(\Delta G+\lambda \right)}^{2}}{4\lambda k_{B}T}\right)$$

Clearly a reasonable correlation between theoretical (3.12 × 10^9^ s^−1^) and experimental rates (2.50 × 10^10^ s^−1^) is observed, with just one order of magnitude difference. The situation is similar for the rate for the charge separation (1.01 × 10^11^ s^−1^ versus 5.9 × 10^11^ s^−1^).

The experimental charge recombination rate to the ground state (7.89 × 10^6^ s^−1^) is strongly underestimated by the Classical Marcus theory. This is likely due to inverted region effects. An overview is given in Table 10.

It has to be noted that in Table 10 we apply two different *V_soc_* values (2.47 cm^−1^ = *CR_T(L)_* and, *V_soc_* value 3.22 cm^−1^, = *CR_T(H)_* is the highest value in Table 2).

Likewise, two different ∆*G* values for *CR*_S_ (−3.08 and −2.87 eV) and for charge separation (CS) from S2 to S1 are applied (−0.178 and −0.49 eV), with either a computational source (e.g., *CR_S(C)_*, see also Figure 14) or based on experimental approximations (e.g., *CR_S(E)_*, see also Table 8). By using computational or experimental energetics, we are able to correlate the Classical Marcus model to the experimental rates.

## 4. Conclusions and Future Outlook

The dependence of the SOCME, as well as $V_{CT}$, as a function of the dihedral angle between the donor and acceptor, have been determined using (TD-)DFT. The opposing effects, maximum at ~90° of the SOCME and minimum of $V_{CT}$ at that same angle, fit perfectly within the current paradigms. At optimized geometries of the locally excited state and singlet charge transfer state (PBE/TZP), values of $V_{SOC}$ = 2.47 cm^−1^ and $V_{CT}=$ 123.2 cm^−1^ are found in NHX. In ACN, the values at the optimized geometries are $V_{SOC}=$ 2.56 cm^−1^ and $V_{CT}=$ 96.4 cm^−1^. The excited state geometries, together with the experimental kinetic data, indicates a relatively large amplitude motion of the donor unit that occurs after photoexcitation (S_2_ → S_1_ → T_1_ in 40 ps). This, together with the time scales, indicate that PyrDMA acts as a molecular “hand wave”, giving a fast push followed by a slow back relaxation (µs timescale triplet decay) to its original location.

The main message of this work, is that the ratios of the optimal rates of charge separation and charge recombination to the triplet, as well as the ratios of the optimal rates for charge recombination to the triplet and charge recombination to the singlet, are governed by their respective couplings, all of which are very sensitive to slight conformational changes. These ratios need to be optimal in order for the SOCT-ISC mechanism to operate efficiently, leading to high triplet yields. Clearly, there are examples of molecular systems in which all these aspects are optimal. The theoretical approach adopted here is clearly relatively simplistic and provides rough estimates of the quantities of interest. More work on the theoretical modeling is needed.

SOCME values for acetone (~62 cm^−1^) and for 4-thio-thymine (~157 cm^−1^) can be reproduced from the literature very well. The SOC values are very dependent on the geometry that is used, but not so much on the XC. LB94 gives the highest values. A long-range parameter of 0.34 (γ) is appropriate for the large range of excited-state geometries, that we studied in this work.

The procedure to use ADF to calculate and predict properties with respect to the SOCT-ISC mechanism should be generalized to other molecules. The ADF-input files of our work could provide a useful starting point for such studies. Future computational studies should emphasize and substantiate the relevance of all aspects for the proposed method. In general, computationally obtained results need to be compared with experimental findings, as exemplified in our study, and this method needs to be optimized where needed [72,73].

## Figures and Tables

**Figure 1 molecules-27-00891-f001:**
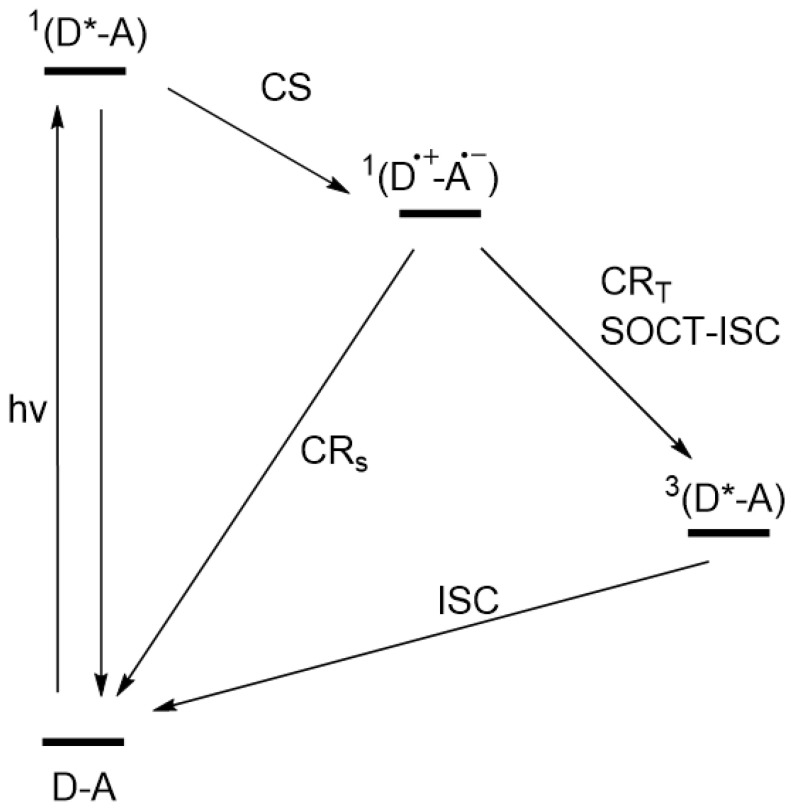
Energy diagram of the photophysical pathways leading to the formation of a triplet state via charge recombination (spin-forbidden CR of ^1^CT to T_1_ (CR_T_)). CR_S_ is the process from ^1^CT to the ground state of the electron donor–acceptor system (D-A). For clarity: ^1^(D*-A) = S_2_, ^1^(D^+.^-A^−.^) = S_1_, ^3^(D*-A) = T_1_. The transition from ^3^(D*-A) back to the ground state is accompanied by an electron spin-flip.

**Figure 2 molecules-27-00891-f002:**
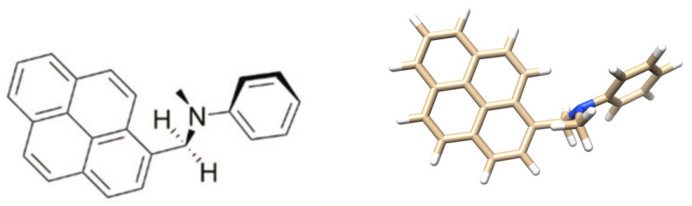
Lewis structure of PyrDMA as well as a 3D representation (Chimera) showing the orthogonality of the donor and acceptor (DFT optimized structure in the ground state).

**Figure 3 molecules-27-00891-f003:**
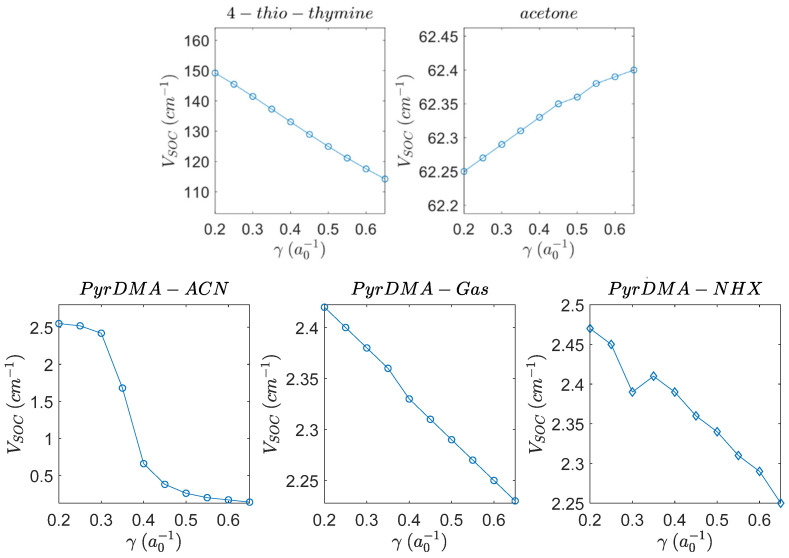
The dependence of $V_{SOC}$ values for 4-thio-thymine, acetone and PyrDMA as a function of the range separation parameter γ (units of inverse Bohr radius = 52.9117 pm) in the gas-phase. Two solvents are also shown for PyrDMA. The geometries are calculated with a TZP basis and PBE XC, while the SOC values are calculated using a TZP basis and the CAMY-B3LYP XC functional. The behavior is similar to the results on acetone and 4-thio-thymine from Gao [3].

**Figure 4 molecules-27-00891-f004:**
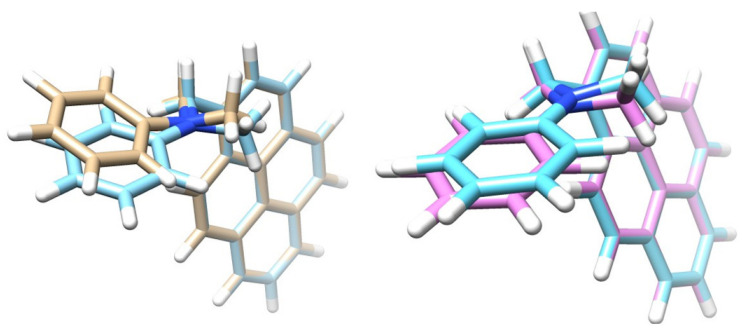
**Left**: A comparison between the optimized molecular structures of the ground state (in blue) and the S_1_ state (in brown). **Right**: A comparison between the S_1_ (=^1^CT, in blue) and the T_2_ (=^3^CT) state (in pink). (PBE/TZP).

**Figure 5 molecules-27-00891-f005:**
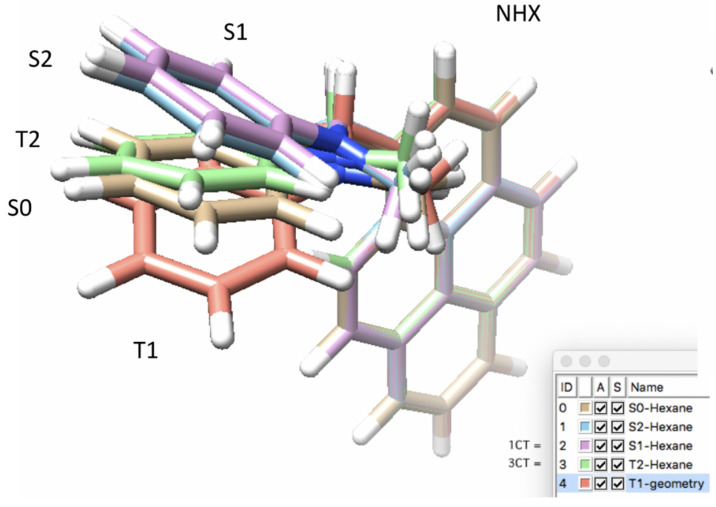
Representation of the five states (PBE/TZP) involved in the photophysical processes of PyrDMA, with overlay of the pyrene unit, clearly showing that the aniline donor group “moves up and down” (relative to the pyrene unit) in going from S_0_, to S_2_, S_1_ and T_1_ in n-hexane solvent.

**Figure 6 molecules-27-00891-f006:**
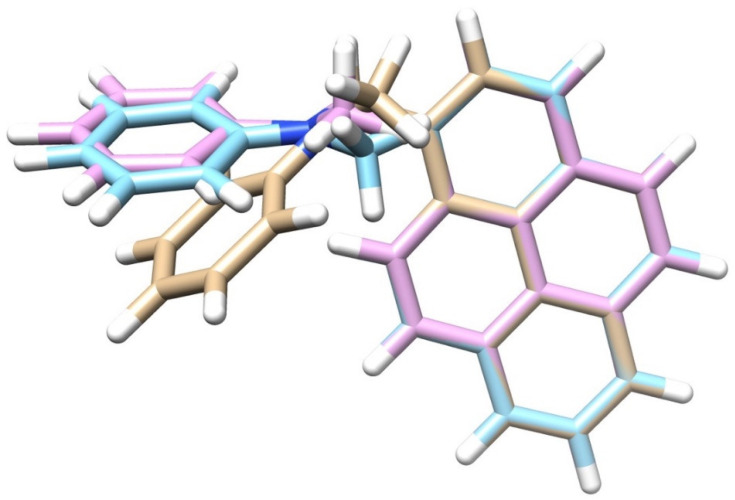
A comparison between the optimized molecular structures of the ^1^CT (=S_1_) states in ACN (in brown) in NHX (in blue) and in the gas-phase (in pink). Note that the molecular structure of ^1^CT is strongly influenced by a polar solvent (CAMY-B3LYP/TZP).

**Figure 7 molecules-27-00891-f007:**
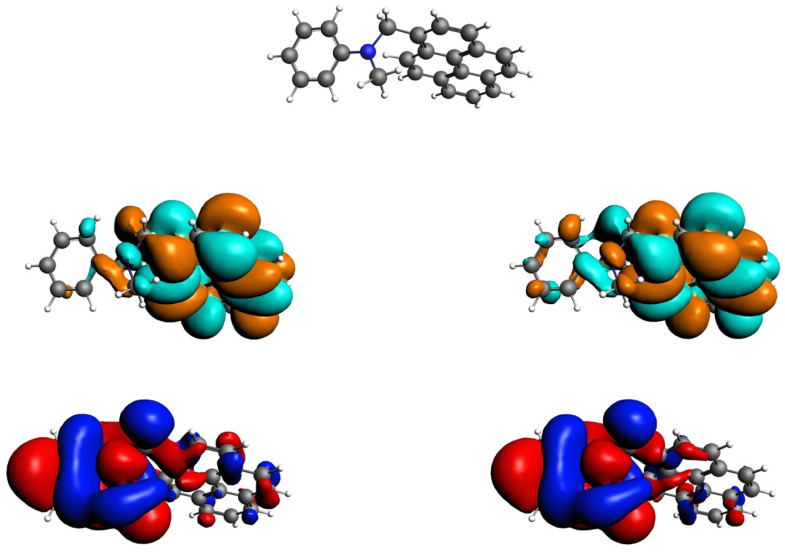
The natural transition orbitals (HONTOs, **bottom** and LUNTOs, **middle**) for the singlet ^1^CT (**left**) and triplet ^3^CT (**right**) charge transfer states at an iso-value of 0.01; (PBE/TZ2P), SS NTO-1, 2.867 eV, S_1_ = ^1^CT and ST NTO-1, 2.854 eV, T_2_ = ^3^CT. The molecular structure, at the same view, is presented at the top. Slight differences can be seen in the orbital extensions to the other unit.

**Figure 8 molecules-27-00891-f008:**
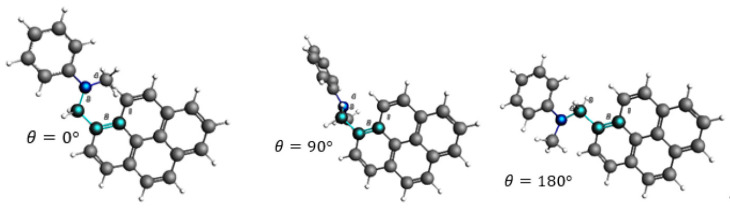
The dihedral angle defined between the pyrene acceptor and the dimethylaniline donor. **Left**: The ground state of PyrDMA at a dihedral angle of 0°. **Centre**: Ground state of PyrDMA at an orthogonal geometry. **Right**: The molecule at a dihedral angle of 180°.

**Figure 9 molecules-27-00891-f009:**
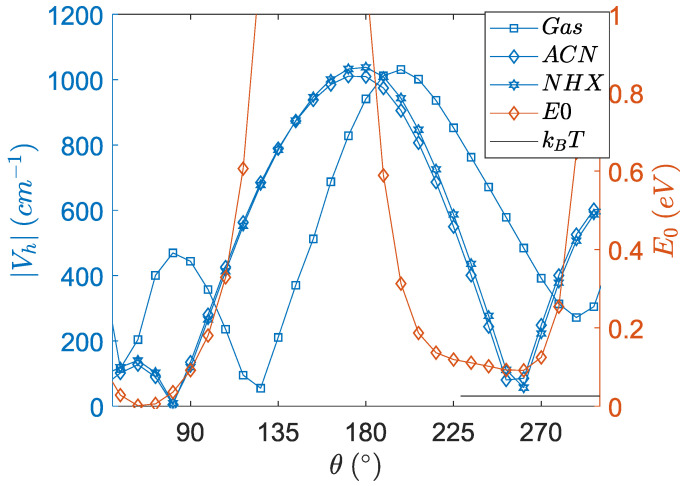
The electronic coupling as function of the dihedral angular coordinate. The calculations are performed starting with the S_2_ geometry in different solvents (PBE/TZP). The electronic coupling is calculated with a TZP basis and the CAM-B3LYP XC functional using the fragment method. The ground state energy (E_0_) in NHX of the intact molecule alongside the thermal energy (*k_B_T)* is plotted to indicate the important range of conformations for the whole intact PyrDMA molecule. The black horizontal line represents the thermal energy available in the ground state. Note that the *x*-axis ranges from 45 to 300 degrees.

**Figure 10 molecules-27-00891-f010:**
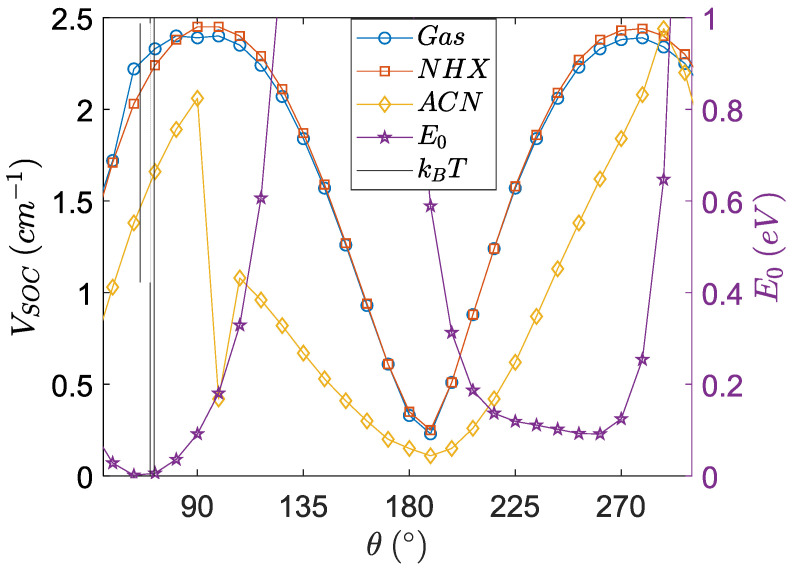
The SOCME values for the singlet charge transfer state ^1^CT as a function of dihedral angle starting with the optimized S_1_ geometry (TZP/PBE) in different solvents for the whole intact PyrDMA molecule. Weak solvent effects are observed for n-hexane and the gas-phase; however, the dependence is modulated in ACN. Black vertical lines indicate the angles at the optimized geometries of the S_1_ state. The quantities are calculated with the TZP basis and the CAMY-B3LYP functional at its default parameters. The ground state energy in NHX (E_0_) of the intact molecule alongside the thermal energy (*k_B_T*) is plotted to indicate the important range of conformations. Note that the *x*-axis ranges from 45 to 300 degrees.

**Figure 11 molecules-27-00891-f011:**
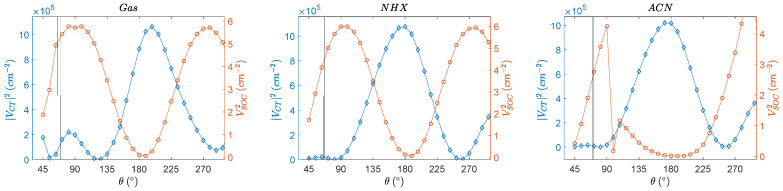
The conformational dependence of the electronic coupling and the SOCME for the singlet charge transfer state. The *V_SOC_* and *V_CT_* curves show complementary behavior. The black vertical lines are the dihedral angles at the optimized S_1_ and S_2_ geometry.

**Figure 12 molecules-27-00891-f012:**
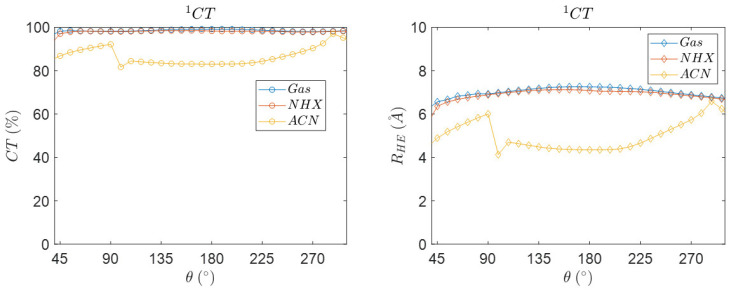
The conformational dependence of the charge transfer character. Charge transfer descriptors for the ^1^CT singlet charge transfer state calculated starting with the S_1_ geometry. The descriptors are calculated as function of the dihedral angle at the optimized singlet charge transfer state S_1_ in different solvents. Geometries are calculated with PBE and TZP. The quantities are calculated with the TZP basis and the CAMY-B3LYP functional.

**Figure 13 molecules-27-00891-f013:**
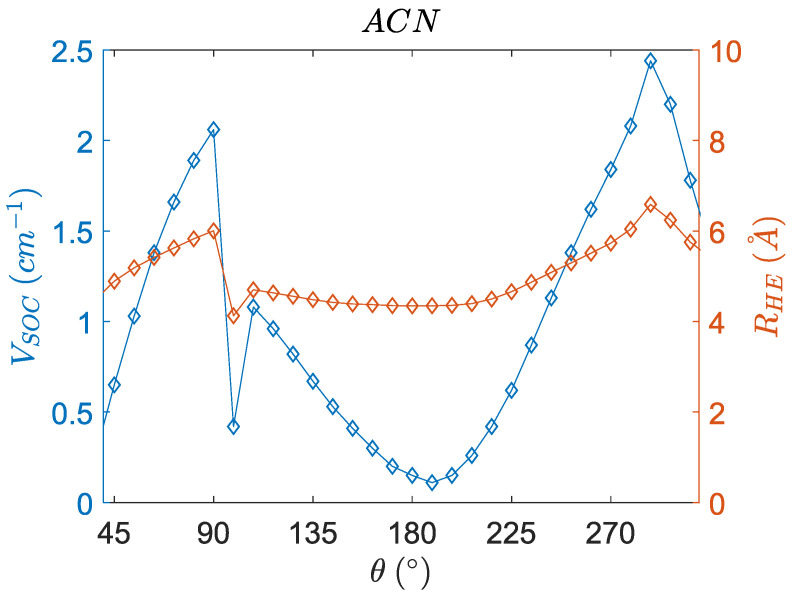
The correlation between the SOCME value and the electron hole distance in ACN. Geometries are calculated with PBE and TZP. The quantities are calculated with the TZP basis and the CAMY-B3LYP functional.

**Figure 14 molecules-27-00891-f014:**
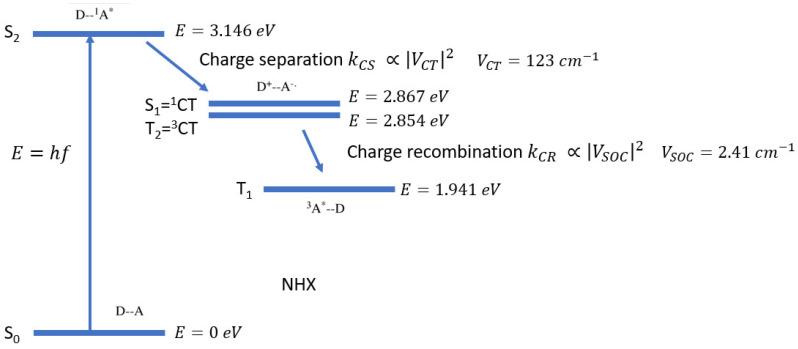
Energies of the excited states, as well as the electronic coupling, and spin–orbit coupling of PyrDMA in n-hexane solvent (NHX), as determined with ADF. The excited state energies are calculated with the TZP basis and PBE functional at the S_1_ geometry, which in return is calculated with the TZ2P basis and PBE functional.

**Table 1 molecules-27-00891-t001:** Experimentally measured rates for charge recombination to the triplet ($k_{CR_{T}})$ in PyrDMA in different solvents [22].

Solvent	$k_{CR_{T}}\;\left(s^{-1}\right)$
NHX	2.5 × 10^10^
DEE	4.3 × 10^9^
ACN	7.0 × 10^8^
NHX *	2.0 × 10^6^

* Intermolecular DMA and Pyrene mixture. Solvents used are n-hexane (NHX), diethyl ether (DEE), acetonitrile (ACN).

**Table 2 molecules-27-00891-t002:** The $V_{SOC}$ values for acetone, 4-thio-thymine and PyrDMA calculated with different exchange correlation functionals at the S_1_ geometry (PBE/TZP).

**XC**	$V_{SOC}\;({\mathrm{cm}}^{-1})$ **Acetone**	$V_{SOC}\;({\mathrm{cm}}^{-1})$ **4-thio-thymine**	$V_{SOC}\;({\mathrm{cm}}^{-1})$ **PyrDMA**		
	Gas	Gas	Gas	NHX	ACN
PW91	61.85	167.50	2.68	2.75	2.85
BLYP	62.56	169.87	2.69	2.76	2.86
CAM-B3LYP	62.20	**156.11**	**2.39**	**2.47**	**2.56**
CAMY-B3LYP	62.30	**138.14**	**2.36**	**2.41**	**1.96**
RPBE	61.00	164.18	2.63	2.69	2.79
HTBS	61.29	166.56	2.66	2.73	2.83
S12y	60.53	160.90	2.59	2.66	2.76
LB94	67.76	181.70	3.15	3.22	3.34
KT1	59.55	156.83	2.59	2.65	2.75
BhandH	60.61	103.83	**2.08**	**2.13**	1.74

XC is the density functional. The geometry optimizations of the S_1_ state were calculated using a TZP basis and PBE XC functional. A TZP basis was used for calculating the SOC. Note the state shift for 4-thio-thymine and the CAM(Y)-B3LYP functionals. Highest values are found for LB94. Values in bold are for S_1_-T_1_ transitions the rest are S_1_-T_2_, (higher value selections).

**Table 3 molecules-27-00891-t003:** The total bonding energies are calculated at the optimized geometries of the states with the TZ2P basis and the CAMY-B3LYP functional with the default parameters for range separation.

Solvent	E_TB_ S_0_ (eV)	E_TB_ S_2_ (eV)	E_TB_ S_1_ (eV)	E_TB_ T_2_ (eV)	E_TB_ T_1_ (eV)
Gas	−389.543	−385.636	−386.151	−386.020	−387.306
NHX	−389.612	−385.839	−385.817	−386.070	−387.375
ACN	−389.700	−386.062	−386.290	−386.153	−387.455

**Table 4 molecules-27-00891-t004:** The differences in total bonding energies are calculated at the optimized geometries of the states with the TZ2P basis and the CAMY-B3LYP functional with the default parameters for range separation, relative to the ground state in the solvent.

Solvent	∆E_TB_ S_0_ (eV)	∆E_TB_ S_2_ (eV)	∆E_TB_ S_1_ (eV)	∆E_TB_ T_2_ (eV)	∆E_TB_ T_1_ (eV)
Gas	0	3.907	3.392	3.523	2.237
NHX	0	3.773	3.795	3.542	2.237
ACN	0	3.638	3.410	3.547	2.245

**Table 5 molecules-27-00891-t005:** The dihedral angle θ of the optimized geometries calculated with the TZ2P basis and the CAMY-B3LYP functional with the default parameters for range separation.

Solvent	S_0_	S_1_	S_2_	T_1_	T_2_
Gas	67.0	69.9	65.7	65.8	64.0
NHX	66.2	65.7	67.1	65.8	63.9
ACN	66.4	71.6	70.4	66.4	64.1

**Table 6 molecules-27-00891-t006:** The spin–orbit and electronic coupling at the optimized geometries. The value of the maximum SOC value near an orthogonal geometry and the maximum electronic coupling near 180°. The geometries are calculated with TZP and the CAMY-B3LYP functional.

	Gas	NHX	ACN
$V_{CT,S2}$ (cm^−1^)	263.11	123.24	96.40
$V_{SOC,S1}$ (cm^−1^)	2.36	2.41	1.95
$\theta_{CT,max}$	197.98	179.99	171.00
$V_{CT,max}$ (cm^−1^)	1031.00	1037.90	1010.70
$\theta_{SOC,max}$	81.06	90.05	89.90
$V_{SOC,max}$ (cm^−1^)	2.40	2.45	2.06

**Table 7 molecules-27-00891-t007:** The center to center distance between pyrene and DMA. The quantity is calculated by taking the average distance between the furthest hydrogen atoms and closest carbon atoms between the donor and acceptor (excluding spacer atoms). The geometries are calculated with the TZ2P basis and CAMY-B3LYP exchange correlation. CRS = charge recombination to the singlet ground state, CRT = charge recombination to the triplet state, CS = charge separation.

Solvent	$R_{C,CS}\;\left(pm\right)$	$R_{C,CRS}\;\left(pm\right)$	$R_{C,CRT}\;\left(pm\right)$
Gas	824.40	801.15	801.15
NHX	827.85	817.65	817.65
ACN	794.00	721.40	721.40

**Table 8 molecules-27-00891-t008:** The Gibbs free energy change for charge separation, the solvent ($\lambda_{s})$ and internal reorganization energies ($\lambda_{i,})$, as well as the solvent properties (n and $\epsilon_{s}$ ). The internal reorganization energies calculated using the four-point method with the TZ2P basis and the CAMY-B3LYP functional. CRS = charge recombination to the singlet ground state, CRT = charge recombination to the triplet state, CS = charge separation.

**Solvent**	n	$\epsilon_{s}$	$\Delta G_{CS}$ (eV)	$\lambda_{s}\;(\mathrm{eV})$	$\lambda_{i,CRS}$ (eV)	$\lambda_{i,CRT}$ (eV)
Gas	1.00	1.00	−0.110	0.00000	0.7817	0.7571
NHX	1.37	1.88	−0.178	0.00135	0.7618	0.7379
ACN	1.34	37.5	−0.493	0.81600	0.7187	0.6965

The following inputs values are used: $r^{-}$ = 3.98 Å, $r^{+}$ = 3.70 Å, $R_{C}$ = 6.50 Å, $E^{0}\left(D^{+}/D\right)=+0.76\;\mathrm{eV}$ vs. SCE in ACN, $E^{0}\left(A/A^{-}\right)=-2.09\;\mathrm{eV}\;\mathrm{vs}.$ SCE in ACN. For pyrene: ^1^$E_{00}=3.26\;\mathrm{eV}$. n is the refractive index of the solvent. $\epsilon_{s}$ is the relative dielectric constant of the medium. $r^{+}$ and $r^{-}$ are the cation and anion radii. SCE is the Saturated Calomel Electrode.

**Table 9 molecules-27-00891-t009:** The solvent reorganization energy calculated from the center to center distance. CRS = charge recombination to the singlet ground state, CRT = charge recombination to the triplet state, CS = charge separation.

Solvent	$\lambda_{s,CS}\;(\mathrm{eV})$	$\lambda_{s,CRS}\;(\mathrm{eV})$	$\lambda_{s,CRT}\;(\mathrm{eV})$
Gas	0.000	0.000	0.000
NHX	0.0018	0.0018	0.0018
ACN	1.0294	0.9326	0.9326

**Table 10 molecules-27-00891-t010:** Parameters for rate determination at T = 295 K in n-hexane (NHX) using the Classical Marcus model.

	∆*G* (eV)	$\lambda_{i}$ (eV)	$\lambda_{s}$ (eV)	*V* (cm^−1^)	*k**_opt_* (s^−1^)	*k**_calc_* (s^−1^)	*k**_exp_* (s^−1^)
*CR_T(L)_*	−0.73	0.7379	0.00135	2.47	1.84 × 10^9^	1.84 × 10^9^	2.50 × 10^10^
*CR_T(H)_*	−0.73	0.7379	0.00135	3.22	3.13 × 10^9^	3.12 × 10^9^	2.50 × 10^10^
*CR_S(C)_*	−3.08	0.7618	0.00135	159	7.84 × 10^12^	2.75 × 10^−22^	7.89 × 10^6^
*CR_S(E)_*	−2.87	0.7618	0.00135	159	7.50 × 10^12^	1.29 × 10^−10^	7.89 × 10^6^
*CS_(C)_*	−0.178	0.7000	0.00135	123.2	4.71 × 10^12^	1.01 × 10^11^	5.9 × 10^11^
*CS_(E)_*	−0.49	0.7000	0.00135	123.2	4.71 × 10^12^	2.52 × 10^12^	5.9 × 10^11^

126.7 ns is the exciplex lifetime in NHX, for intermolecular interaction, [71] the following energy values were taken ^1^E_00_ = 3.26 eV, ^1^E_CT_ = 2.87 eV, ^3^E_00_ = 2.14 eV, next to the values in Table 6 and Table 8. We apply two different *V_soc_* values (2.47 cm^−1^ = *CR_T(L)_* and *V_soc_* value 3.22 cm^−1^, = *CR_T(H)_*); we use two different ∆*G* values with either computational source (e.g., *CR_S(C)_*) or based on experimental approximations (e.g., *CR_S(E)_*).

## Data Availability

In relation to Research Data Management, Appendix A is also available here: https://doi.org/10.21942/uva.17198408.v1 (accessed on 25 January 2022).

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
