# Peer review of "Spin Orbit Coupling in Orthogonal Charge Transfer States: (TD-)DFT of Pyrene—Dimethylaniline"

_molecules, 2022, doi:10.3390/molecules27030891_

Round 1
Reviewer 1 Report
The manuscript by Bissesar et al. aims to characterize, through theoretical first principles approaches based on Density Functional Theory (DFT) and time-dependent DFT, some aspects related to the spin-orbit charge transfer intersystem crossing (SOCT-ISC) mechanism, and eventually identify which ones favor such mechanism more. In particular, the N-methyl-N-phenyl-1-pyrene-methanamine (PyrDMA) prototypical molecule is considered, and the simulations are performed via the Amsterdam Density Functional (ADF) computer code. The Authors obtain the optimized ground and selected excited state molecular geometries in gas-phase, n-hexane (NHX), and acetonitrile (ACN) solvents at the PBE/TZP level of theory. They compute the spin-orbit coupling (SOC) matrix element (ME) and charge transfer (CT) integral employing the results of the (TD)DFT simulations, possibly according to Ref. 40 [Zhu et al., Spectrochim. Acta - Part A Mol. Biomol. Spectrosc. 221, 117214 (2019)], for instance, and benchmarking a few exchange-correlation (XC) functionals, as well as the screening factor of the range separation function required by the range-separated functionals, for the single-points (TD)DFT simulations. The most important and, to the best of my knowledge, original contribution (regarding the molecule considered by the Authors) is the estimation of SOCME, CT integral and character, and electron-hole distance as a function of the molecular dihedral angle. Ultimately, the evaluation of Gibbs free energy for change separation, reorganisation energies, and charge separation and recombination rates is attempted.
The manuscript is not well written and it is often difficult to understand what the Authors mean. Concepts and results are often re-stated in different section making it difficult to follow what the Author did, and ultimately impossible in Section 3.6. I believe the manuscript needs some major work on the text.
Part of the conclusions are not actually supported by the results shown in the manuscript. In particular, I don't believe Table 10 can lead to any of the conclusions the Authors state in the text. This may be partially related to the confusing way Section 3.6 is written.
Nonetheless, part of the work, such as the one on the dihedral angular dependence of the quantities computed from the ab initio simulations, is (to the best of my knowledge) original and could be of interest to some of the readers in the community. Therefore, my suggestion is to consider the manuscript for publication in Molecules, after the Authors have comprehensively and carefully addressed the major comments detailed in some points below.
1. As mentioned earlier Section 3.6 is very difficult to understand and should be made more clear.
2. Lines 542-546. It is not clear how this is relevant for the system studied in this work and how relevant is the experimental evaluation mentioned.
3. Lines 547-550. I do not understand this. Does it mean that the related analysis is inconclusive?
4. Lines 551-570 are misleading. I do not see the correlation mentioned by the Authors between computed and measured values reported in Tab.10. While I understand most of the rationalisation and reasoning up to Section 3.5, I do believe the values reported in Tab.10 do not allow to draw the conclusion reported here and in Section 4. In particular, lines 578-583 and 599-600 are not substantiated by the results presented in the manuscript in the current form. Also, what is the difference between the lines identified by the same letters in the first column of Tab.10?
5. Lines 601-602. The procedure described here looks very system-dependent, how do the Authors propose to generalise it to "other molecules"?
6. Lines 604-605. Can the Author indicate where the approach adopted needs to be optimized and why the comparison to experimental findings is not possible at the moment?
7. Line 139-142: there is no such thing as a "default SCF convergence criterion" as it is always system-dependent, the convergence should be checked with respect to the simulated electronic structure and the properties of interest, and then a convergence criterion set up a posteriori, regardless of any default option available in computational codes. The basis set is critical for convergence as well, did the Authors benchmark it too? Also what is the unit of the number in line 140? The "Lisa cluster of SURFsara" would need a reference, as well as the SPARTAN code.
8. It is not clear if the Authors implemented the approach they use in ADF or if they used the code to run the (TD)DFT simulations from which, then, they extract the quantities from which they compute the SOCME, etc. It is also not clear how they compute the SOCME from the ADF (TD)DFT results, as Eq.(2) refers to the SO hamiltonian in Eq.(1), which does not relate to standard quantities routinely obtained from ab initio simulations.
9. Lines 169-174: I believe the SOCME values shown in Tab.2 are computed by the Authors, but from the text it seems like they refer to the values computed in Ref.[3] for acetone and 4-thio-thymine, could the Authors clarify? Are there experimental values to compare with (lines 193-194)?
10. Lines 175-177: I expect the ground state (GS) and the excite state geometries differ substantially, in general, and the choice of the geometry should be dictated by the states involved in the SOCME of interest. Also, why only a few values of Tab2 refer to the S1-T1 transition, while most are for the S1-T2? Isn't that something that can be controlled during the simulation? Moreover, the "waving hand" behavior (line 246) was expected, given the molecular structure and the fact that the pyrene unit's geometry remains, again as expected, practically unchanged after excitation.
11. Related to the previous point, I understand the Authors perform the geometry optimizations of the GS and the excite state geometries at the PBE/TZP level and then use these optimized geometries for the single point simulations when benchmarking different XC functionals for the SOCME calculations. This way, however, the effect different XC functionals may have on the optimized structure, which for this molecule I would expect to be significant, is not accounted for. Can the Author explain why did they neglect it? Also, while I understand why CAM-B3LYP or CAMY-B3LYP should be the XC functionals of choice in this case, based on the literature, I do not understand why CAMY-B3LYP is chosen after the results shown in Tab.2. The Authors state in the conclusions that "CAM-B3LYP is more suited to calculate properties at single point geometries" (lines 597-598) but in the nest sentence they seem to doubt such conclusion, on the other hand CAMY-B3LYP is used throughout the work, can the Authors comment further?
12. I did not get how an optimal value of 0.34 was chosen for the screening factor of the range separation function (parameter "gamma") based on the benchmark shown in Fig.3. Also, the top-right and bottom-right plots in Fig.3 are the same.
13. The solvent seems to critically affect the results presented in the manuscript, especially in the case of ACN solvent. How did the Authors perform the simulations of the PyrDMA molecule in different solvents with ADF? Did they adopt an implicit or explicit solvent approach? Is the solvent approach able to properly describe highly polar solvents?
14. Lines 500-503. This is expected, given the much higher polar character and dielectric constant of the ACN solvent. Can the Author comment further?
15. While I see how the dihedral angles reported in Tab.5 are useful to describe the GS and excited states geometries and to interpret the SOCME results, I do not see why the total bonding energies reported in Tab.3 and Tab.4 should be. Could the Authors comment?
16. Lines 284-298: The excitation energies computed by the Authors are not shown and it is not clear how well they compare with available data. Excited states energy differences are also relevant when establishing the efficiency of the charge separation and recombination channels.
17. Regarding Figs.9-13. The reason for the strange behavior of SOCME and other quantities in ACN solvent around a dihedral ange of about 100° should be clarified. Also, why is the range 45-300°? There are no black vertical lines in Fig.9 (lines 330-331). Line 337, "Around 0° and 360°" is out of the range shown in the plots. Why is the ground state energy (E0) only shown for NHX solvent?
18. Why did the Authors use the TZP basis set for the calculations of SOCME and other quantities (Figs.9-13), while they use a TZ2P basis set for the calculations of center to center distance between pyrene and DMA units (Tabs.7,8)? Are the results consistent? Same for the excited state energies in Fig.14. Also, what is the difference between the third and fourth columns in Tab.7 and Tab.9?
19. Lines 464-470. Do the Authors have an estimate of the effect the choice of donor and acceptor parts of the molecule have on the calculations and analysis of the reorganization energies?
20. Lines 136-137, what do the Authors mean when they say the "computational results ... are presented when available"?
21. Lines 473-476. How were these values computed, i.e., which approach, XC, basis, computational parameters? Would a comparison with similar results obtained from the simulations performed by the Authors be meaningful?
Typos and others:
- The format of the references should be uniform throughout the text ( [ ] instead of apex);
- Abbreviations should be defined once and used in place of the "extended version", going back and forth and re-defining abbreviations throughout the manuscript is misleading for the reader;
- Line 83: "play" --> "plays";
- Line 91: "Buck et al." --> "Buck et al. [26]";
- Line 114: "minimized" --> "optimized";
- Line 169: "basis sets and" --> "basis sets";
- Line 243: "structure" --> "structures";
- Line 322: "and hydrogen atoms are added to the free valences", the Authors probably mean that "the dangling bonds are passivated with H atoms";
- Line 374: "state as" --> "as";
- Line 380: "current scientific paradigm", do the Authors mean current understanding of the mechanism?
- Lines 404-407: in the plot the Authors used 0-100% instead of 0-1;
- What does "SCE" stand for in the caption of Tab.8?
- Line 451: delete "an";
- Line 452: "component" --> "components";
- Line 471: "Table" --> "in Table";
- Line 489: delete "out";
- Lines 479-480 should be placed earlier in the text, as they illustrate some parameters used also in Tab.8;
- Lines 483-484: the sentence does not make sense.
- Line 496: delete "in polar solvents";
- Please provide a reference for Eq.(10);
- Line 566: "Classsical" --> "Classical";
- Line 587: "which all" --> "all of which";
Author Response
Referee 1.
We thank the referee for the extensive and very useful comments.
We repeat the comments here (in blue) and our response to every comment:
- As mentioned earlier Section 3.6 is very difficult to understand and should be made more clear.
1
We have improved to the best of our abilities the section 3.6.
The referee is right that this section is complex. We have tried to make it more clear an access-able. We have adapted the text and deleted some inconsistencies. A native speaker has improved the text.
- Lines 542-546. It is not clear how this is relevant for the system studied in this work and how relevant is the experimental evaluation mentioned.
2
We have adapted Lines 542-546 showing more clearly the importance and relation of this work with our manuscript. They now read:
“Interestingly, for a molecule very similar to PyrDMA, electron transfer parameters have been reported, using a very different (multi-parameter fitting) approach. For a ‘donor-inversed’ molecule (P1D), in which the amino unit is para relative to the CH2 linker such parameters have been estimated (= 0.543 eV; S = 0.65, = 0.840 eV, = 6775 cm-1).[70] The Gibbs free energy for electron transfer in acetonitrile was evaluated experimentally to be -0.48 eV, and is also reported in this work.”
- Lines 547-550. I do not understand this. Does it mean that the related analysis is inconclusive?
3
We have adapted Lines 547-550 to make it more clear, which now read:
“However, so far we were unable to make a good determination of S with DFT for PyrDMA. The S values obtained for PyrDMA so-far were very low (too low, S~0.001), which we assume to be due to the large conformational changes between the charge transfer state and the triplet state. Based on the couplings, the energetic and kinetic experimental data we estimate the S values for PyrDMA to be ~7, ~5, ~1 for respectively CS, CRS and CRT (data not shown). Unfortunately, we cannot approach the rates with the Semi-Classical Marcus model.”
- Lines 551-570 are misleading. I do not see the correlation mentioned by the Authors between computed and measured values reported in Tab.10. While I understand most of the rationalisation and reasoning up to Section 3.5, I do believe the values reported in Tab.10 do not allow to draw the conclusion reported here and in Section 4. In particular, lines 578-583 and 599-600 are not substantiated by the results presented in the manuscript in the current form. Also, what is the difference between the lines identified by the same letters in the first column of Tab.10?
4
We have adapted the section of Lines 551-570 . We have made changes to the lines 578-583 and 599-600.
We have inserted extra information in table 10 to make more clear what we did.
It now reads:
“Clearly a reasonable correlation between theoretical (3.12 ´ 109 s-1) and experimental rates (2.50 ´ 1010 s-1 ) is observed, with just one order of magnitude difference. The situation is similar for the rate for the charge separation (1.01 ´ 1011 s-1 versus 5.9 ´ 1011 s-1).
The experimental charge recombination rate to the ground state (7.89 ´ 106 s-1) is strongly underestimated by the Classical Marcus theory. This is likely due to inverted region effects. An overview is given in Table 10.
It has to be noted that in Table 10 we apply two different Vsoc values (2.47 cm-1 = CRT(L) and Vsoc value 3.22 cm-1, = CRT(H) the highest value in Table 2).
Likewise, two different DG values for CRS (-3.08 and -2.87 eV) and for charge separation (CS) from S2 to S1, (-0.178 and -0.49 eV), with either computational source (e.g. CRS(C), see also figure 14) or based on experimental approximations (e.g. CRS(E), see also Table 8) are applied. By using computational or experimental energetics we are able to correlate the Classical Marcus model to the experimental rates.”
Similar notes are also in the caption of table 8 now.
- Lines 601-602. The procedure described here looks very system-dependent, how do the Authors propose to generalise it to "other molecules"?
5.
We have inserted: “The ADF-input files of our work could provide a useful starting point for such studies”.
- Lines 604-605. Can the Author indicate where the approach adopted needs to be optimized and why the comparison to experimental findings is not possible at the moment?
We adapted this section to:
“In general, computationally obtained results need to be compared with experimental findings, as exemplified in our study, and this method needs to be optimized where needed”
- Line 139-142: there is no such thing as a "default SCF convergence criterion" as it is always system-dependent, the convergence should be checked with respect to the simulated electronic structure and the properties of interest, and then a convergence criterion set up a posteriori, regardless of any default option available in computational codes. The basis set is critical for convergence as well, did the Authors benchmark it too? Also what is the unit of the number in line 140? The "Lisa cluster of SURFsara" would need a reference, as well as the SPARTAN code.
7
The convergence criteria were the same as used in a previous DFT study (Inorg. Chem. 2020, 59, 1496−1512). We specify the convergence criterion, also in the input files. The unit is energy per volume unit. We believe it is eV/Å3. We refer to the ADF developers for verification.
We inserted the requested references.
SurfSARA is in the Acknowledgements:
“Acknowledgments: This project has received funding from the European Union’s Horizon 2020 research and innovation programme under the Marie Sklodowska-Curie grant agreement no. 764837. We thank the Universiteit van Amsterdam for structural support. This work was carried out on the Dutch national e-infrastructure with the support of SURF Cooperative.”
- It is not clear if the Authors implemented the approach they use in ADF or if they used the code to run the (TD)DFT simulations from which, then, they extract the quantities from which they compute the SOCME, etc. It is also not clear how they compute the SOCME from the ADF (TD)DFT results, as Eq.(2) refers to the SO hamiltonian in Eq.(1), which does not relate to standard quantities routinely obtained from ab initio simulations.
8.
We have inserted the information about the SOPERT keyword in ADF.
This hopefully makes it more clear that we used ADF as a tool.
- Lines 169-174: I believe the SOCME values shown in Tab.2 are computed by the Authors, but from the text it seems like they refer to the values computed in Ref.[3] for acetone and 4-thio-thymine, could the Authors clarify? Are there experimental values to compare with (lines 193-194)?
9
We have adapted this section to make it more clear. It now reads:
“The first step in this work was therefore to benchmark the method for the SOC matrix () determination, for which acetone and 4-thio-thymine were used as reference molecules.[3] The SOCME () for these molecules was calculated within our work with different exchange correlation functionals (see Table 2). Using the ground state geometry results in much lower values for . Therefore, the calculations were done at the optimized excited state S1 geometry (PBE exchange correlation/TZP basis). The SOCME values are benchmarked with ADF using different exchange functionals. The SOC matrices that we obtain correlate well with those previously reported in a benchmark study.[3]”
And:
“It can be noted that we do this benchmark study relative to the reported work[3] on other computational methods. We do not claim comparison to precise experimental data, which is unknown to us.”
To the best of our knowledge there is no experimental information of the SOCME for acetone nor for thiothymine.
- Lines 175-177: I expect the ground state (GS) and the excite state geometries differ substantially, in general, and the choice of the geometry should be dictated by the states involved in the SOCME of interest. Also, why only a few values of Tab2 refer to the S1-T1 transition, while most are for the S1-T2? Isn't that something that can be controlled during the simulation? Moreover, the "waving hand" behavior (line 246) was expected, given the molecular structure and the fact that the pyrene unit's geometry remains, again as expected, practically unchanged after excitation.
10.
We have inserted more information at table 2 about the selection of the values.
We are surprised that the ‘waiving hand’ is expected by the referee. We expected very little changes in the excited state geometries and are still surprised by the large conformational difference of the triplet state relative to the other states.
- Related to the previous point, I understand the Authors perform the geometry optimizations of the GS and the excite state geometries at the PBE/TZP level and then use these optimized geometries for the single point simulations when benchmarking different XC functionals for the SOCME calculations. This way, however, the effect different XC functionals may have on the optimized structure, which for this molecule I would expect to be significant, is not accounted for. Can the Author explain why did they neglect it? Also, while I understand why CAM-B3LYP or CAMY-B3LYP should be the XC functionals of choice in this case, based on the literature, I do not understand why CAMY-B3LYP is chosen after the results shown in Tab.2. The Authors state in the conclusions that "CAM-B3LYP is more suited to calculate properties at single point geometries" (lines 597-598) but in the nest sentence they seem to doubt such conclusion, on the other hand CAMY-B3LYP is used throughout the work, can the Authors comment further?
11.
We have deleted the statement that caused the confusion. We used the best methods that were available and that we could extract from literature.
- I did not get how an optimal value of 0.34 was chosen for the screening factor of the range separation function (parameter "gamma") based on the benchmark shown in Fig.3. Also, the top-right and bottom-right plots in Fig.3 are the same.
12.
The gamma value of 0.34 seem as a good starting point, judging from the graphs in figure 3.
It should not be much higher. But is still low enough to give reasonable values in all solvents.
We corrected Figure 3. It indeed contained a mistake and we thank the referee for noting this.
- The solvent seems to critically affect the results presented in the manuscript, especially in the case of ACN solvent. How did the Authors perform the simulations of the PyrDMA molecule in different solvents with ADF? Did they adopt an implicit or explicit solvent approach? Is the solvent approach able to properly describe highly polar solvents?
13.
We specified more clearly that we used the COSMO method, in the experimental part.
It may be too simplistic, but for the main solvent of interest, n-hexane, it seems reasonable.
- Lines 500-503. This is expected, given the much higher polar character and dielectric constant of the ACN solvent. Can the Author comment further?
14.
We have no further comments on the Delta G value and solvent polarity.
- While I see how the dihedral angles reported in Tab.5 are useful to describe the GS and excited states geometries and to interpret the SOCME results, I do not see why the total bonding energies reported in Tab.3 and Tab.4 should be. Could the Authors comment?
The total bonding energies are given as extra information and indeed do not play a role in the SOCT-ISC mechanism.
- Lines 284-298: The excitation energies computed by the Authors are not shown and it is not clear how well they compare with available data. Excited states energy differences are also relevant when establishing the efficiency of the charge separation and recombination channels.
16.
We now indicate more cleary that the excitation energies are reported in figure 14.
“The excitation energies increase in the order S0,T1,T2, S1 and S2 (see also figure 14).”
- Regarding Figs.9-13. The reason for the strange behavior of SOCME and other quantities in ACN solvent around a dihedral ange of about 100° should be clarified. Also, why is the range 45-300°? There are no black vertical lines in Fig.9 (lines 330-331). Line 337, "Around 0° and 360°" is out of the range shown in the plots. Why is the ground state energy (E0) only shown for NHX solvent?
17.
We agree with the referee that this is a very interesting effect. But it is not the focus of our study, and we specify this in the text now more clearly. The SOCT ISC mechanisms is important in n-hexane solvent for this molecule. We now specify that the reason is clarity for the choice of angles. We indicate that the other curves are in the supporting information.
- Why did the Authors use the TZP basis set for the calculations of SOCME and other quantities (Figs.9-13), while they use a TZ2P basis set for the calculations of center to center distance between pyrene and DMA units (Tabs.7,8)? Are the results consistent? Same for the excited state energies in Fig.14. Also, what is the difference between the third and fourth columns in Tab.7 and Tab.9?
We have inserted more information on table 7 and 9 in the captions of the tables.
We inserted a comment in the experimental on consistency on the basis sets.
- Lines 464-470. Do the Authors have an estimate of the effect the choice of donor and acceptor parts of the molecule have on the calculations and analysis of the reorganization energies?
19.
We do not have further information on the choice of donor or acceptor parts and reorganization energies.
- Lines 136-137, what do the Authors mean when they say the "computational results ... are presented when available"?
We now write:
“However, computational results regarding the 3CT are presented in our manuscript when appropriate.”
We hope that this is now more clear.
- Lines 473-476. How were these values computed, i.e., which approach, XC, basis, computational parameters? Would a comparison with similar results obtained from the simulations performed by the Authors be meaningful?
21.
We refer to references 61 and 62 for more information on the methods used in these papers. The outcome shows that our values are a bit on the high side.
We have adapted the Typos and others errors.
We thank especially this referee for the large efforts to help us improve this work.

Reviewer 2 Report
The manuscript reports on investigating the dependence of the SOC matrix element and the electronic coupling for charge separation on the example of a pyrene-acceptor and a dimethylaniline donor system. This work is a part of a series of works that provide additional clues about organic compounds' charge transfer processes. For example, such theoretical investigations are essential in understanding and predicting temperature-activated delayed fluorescence (based on Singlet-triplet reversible ISC), which could be observed on the organic compound possessing CT processes. There is a lot of data making up the whole picture of the investigation. The calculation experiments seem correct and adequate, and the data support conclusions. Therefore, I think the paper could accept after minor revisions:
- Figure 1. What does it mean pathway 3(D*-A) to the ground state D-A named as ISC process? Usually, the ISC is observed for excited singlet-triplet conversion. Correct or add additional explanations.
- Figure 7. According to the picture, even GS demonstrates the different symmetry of orbitals (coloring of donor fragment is different). It is also could be assigned to the excited singlet and triplet states. For better understanding, I suggest dividing the picture into two fragments: at left HONTO at right LUNTO for singlet and triplet excitations with similar color codes blue-red for example.
Author Response
Referee 2
We thank the referee for the useful comments.
We repeat the comments here in blue:
Figure 1. What does it mean pathway 3(D*-A) to the ground state D-A named as ISC process? Usually, the ISC is observed for excited singlet- triplet conversion. Correct or add additional explanations.
Figure 7. According to the picture, even GS demonstrates the different symmetry of orbitals (coloring of donor fragment is different). It is also could be assigned to the excited singlet and triplet states. For better understanding, I suggest dividing the picture into two fragments: at left HONTO at right LUNTO for singlet and triplet excitations with similar color codes blue-red for example.
We have inserted an extra sentence in the caption of figure 1, for clarity:
“The transition from 3(D*-A) back to the ground state is accompanied by an electron spin flip.”
We have adjusted figure 7 as suggested by the referee showing the orbitals in separate pictures.
We thank the referee for the efforts to help us improve this work.

Reviewer 3 Report
In this manuscript, Williams and co-workers evaluate the matrix elements of spin-orbit coupling for a charge recombination process in a selected electron donor-acceptor molecule, PyrDMA. The article is well written, and the topic is timely. The results are relevant, and the computational level used is correct. The authors have experience in the field, and there is no major critique. Consequently, I recommend this manuscript for publication in Molecules, and only minor aspects should be improved and/or clarified.
In the conclusions, the authors indicate that “the impression is that PBE is more appropriate for geometry optimization (leading to less extreme conformations) and that CAM-B3LYP is more suited to calculate properties at single point geometries”. In this work, the geometry effect of using different functional methods has not been explored (the optimizations have been performed at the PBE/TZP level). Then, some references should be added to reinforce this sentence or additional details should be provided.
The authors should include references of the density functional theory methods used in the work. In addition, some TD-DFT references should be included.
In the list of abbreviations, it should be added diethyl ether, DEE.
Author Response
Referee 3.
We thank the referee for the useful comments.
We repeat the comments here in blue:
“In the conclusions, the authors indicate that “the impression is that PBE is more appropriate for geometry optimization (leading to less extreme conformations) and that CAM-B3LYP is more suited to calculate properties at single point geometries”. In this work, the geometry effect of using different functional methods has not been explored (the optimizations have been performed at the PBE/TZP level). Then, some references should be added to reinforce this sentence or additional details should be provided.
The authors should include references of the density functional theory methods used in the work. In addition, some TD-DFT references should be included.
In the list of abbreviations, it should be added diethyl ether, DEE.”
We have removed the sentence
“the impression is that PBE is more appropriate for geometry optimization (leading to less extreme conformations) and that CAM-B3LYP is more suited to calculate properties at single point geometries”.
from the conclusions.
The referee is right that we can not substantiate this statement.
We here compile the references on DFT and TD-DFT:
References related the development and application of (TD-)DFT are, in the revised version:
25, 29, 35, 36, 37, 40, 41, 42, 45, 52, 55, 57, 58, 61, 69.
We have added the abbreviation of DEE in the list.
We thank the referee for the efforts to help us improve this work.

Round 2
Reviewer 1 Report
I went through the cover letter in response to the referees' report and new version of the manuscript by Bissesar et al. The manuscript has now improved, albeit not dramatically, from the original version and is quite clearer. The Authors' reply to the referee's report is sound and the edits introduced as a result of the point-by-point comments are appropriate, although in some cases a bit hasty.
I still think the results shown in Tab. 10 do not support completely the conclusions, but, now that the reasoning is explained slightly better, it seems they do not contradict the Authors' conclusion either. Theoretical and experimental rates are at least one order of magnitude off. The theoretical approach is clearly a bit simplistic and provides rough estimates at best of the quantities the Authors are interested in, more work on the theoretical modelling is needed. This has to be clearly stated in the conclusion. Nevertheless, my suggestion is to consider the manuscript for publication in Molecules, after the Authors have addressed the comments detailed in some points below.
1. As mentioned, the theoretical approach adopted is clearly a bit simplistic and provides rough estimates of the quantities the Authors are interested in, more work on the theoretical modelling is needed. This has to be stated in the conclusions.
2. I think a comment regarding the first part of point 11 of my previous referee report would benefit the manuscript and could be added: "11. Related to the previous point, I understand the Authors perform the geometry optimizations of the GS and the excite state geometries at the PBE/TZP level and then use these optimized geometries for the single point simulations when benchmarking different XC functionals for the SOCME calculations. This way, however, the effect different XC functionals may have on the optimized structure, which for this molecule I would expect to be significant, is not accounted for. Can the Author explain why did they neglect it?"
3. The authors should add a comment on the fact that the approach to describe the solvent may be too simplistic to correctly model ACN and therefore the peculiar results they get with such solvent may not be completely reliable.
4. Why only a few values of Tab2 refer to the S1-T1 transition, while most are for the S1-T2? Isn't that something that can be controlled during the simulation?
